# Three Varieties of Grape Pomace, with Distinctive Extractable:Non-Extractable Polyphenol Ratios, Differentially Reduce Obesity and Its Complications in Rats Fed a High-Fat High-Fructose Diet

**DOI:** 10.3390/foods12071370

**Published:** 2023-03-23

**Authors:** Yuridia Martínez-Meza, Alexandro Escobar-Ortiz, Fernando Buergo-Martínez, Iza Fernanda Pérez-Ramírez, Jara Pérez-Jiménez, Luis M. Salgado, Rosalía Reynoso-Camacho

**Affiliations:** 1Facultad de Química, Universidad Autónoma de Querétaro, Querétaro 76010, Qro., Mexico; 2Department of Metabolism and Nutrition, Institute of Food Science, Technology and Nutrition (ICTAN-CSIC), José Antonio Novais 10, 28040 Madrid, Spain; 3CIBER of Diabetes and Associated Metabolic Disease (CIBERDEM), Instituto de Salud Carlos III, 28029 Madrid, Spain; 4CICATA-Querétaro, Instituto Politécnico Nacional, Querétaro 76010, Qro., Mexico

**Keywords:** grape pomace, obesity, extractable polyphenols, non-extractable polyphenols

## Abstract

Grape pomace is a commonly discarded by-product characterized by high extractable (EPP) and non-extractable (NEPP) polyphenol contents which exhibits anti-obesogenic effects. However, the relevance of each fraction needs to be elucidated. In this work, we examined the effects of three pomaces with different concentrations of EPPs and NEPPs on metabolic alterations associated with obesity. The NEPP:EPP ratio of the grape pomaces was 1.48 for Malbec, 1.10 for Garnacha, and 5.76 for Syrah grape varieties. Rats fed a high-fat high-fructose diet supplemented with Malbec grape pomace (HFFD + MAL) Syrah grape pomace (HFFD + SYR) or Garnacha grape pomace (HFFD + GAR) showed significantly less weight gain: 20%, 15%, and 12% less, respectively, compared to HFFD controls. The adiposity index was also significantly decreased by 20% in the HFFD + MAL and HFFD + SYR groups, and by 13% in the HFFD + GAR group. Serum triglycerides were significantly decreased by 46% in the HFFD + MAL group and by 31% in the HFFD + GAR group, compared to the HFFD group, but not in the HFFD + SYR group. All pomace supplementations regulated postprandial glucose in an oral glucose tolerance test. Therefore, grape pomaces containing both EPPs and NEPPs exert beneficial effects on body weight and glucose homeostasis, while EPPs seem to control triglyceride levels more effectively.

## 1. Introduction

Grape pomace is a by-product of winemaking and source of bioactive phenolic compounds [1]. Several studies have evaluated the effects of extracts from grape pomace consisting of fractions rich in extractable polyphenols (EPPs). However, grape pomace is also an important source of non-extractable polyphenols (NEPPs), which are not obtained by common extraction procedures due to their high molecular weight, complex structure, and the fact that they tend to bind to macromolecules such as carbohydrates and proteins, thus remaining in the food matrix after conventional extractions [2]. According to Martins et al. [3], the extraction and purification of NEPPs require the application of hydrolysis (alkaline or acid) or enzymatic methods, but these modify the original NEPP structures. Moreover, if a specific EPP-rich or NEPP-rich extract was obtained from grape pomace, then another unused by-product would be left and the potential complementary biological effects of EPPs and NEPPs working in tandem would not be exploited. Instead, using the complete matrix of the grape pomace, without alterations, represents a low-cost alternative with high nutraceutical potential [4].

Obesity has become a major public health problem, defined by many authors as a pandemic, with several associated pathologies [5]. Targeting this pathology involves multiple strategies, including exploring food ingredients able to modify insulin resistance [6] as well as the circulating levels of triglycerides, as the main fat deposits, solely contributing to insulin resistance [7]. Some of these effects have been associated to polyphenols [8,9]. However, due to the complexity of the extraction and purification of NEPPs, it has been a challenge to associate these specific compounds with health benefits. The majority of studies on the particular health benefits of grape pomace have focused on extracts containing mostly EPPs. In this regard, it has been demonstrated that EPPs decrease mass gain of fat, improve glucose tolerance [10], and reduce adipocyte diameter [11]. Furthermore, (+)-catechin and (−)-epicatechin, compounds found in EPP extracts from grape pomace, have been associated with decreased gain of body weight and adipose tissues in mice [12]. In addition, some authors have attributed the beneficial effects of grape pomace to proanthocyanidins, including decreased body weight and white adipose tissue relative weight, and improved lipid metabolism and inflammation status [13]. However, it should be highlighted that whole grape pomace, in which the EPP concentration is not as high as in EPP extracts, also exerts beneficial biological effects. For instance, rats and mice fed a hypercaloric diet and supplemented with grape pomace showed reduced serum glucose and triglycerides [14], decreased body weight gain and ectopic fat deposition, and improved glucose tolerance [15]. Moreover, in adults with at least one or two components of metabolic syndrome, supplementation with grape pomace showed improved glucose tolerance which was attributed to its ability to improve insulin sensitivity [16,17].

Previous scientific studies of whole grape pomace raise the question of the contribution of NEPPs to the biological effects of this material since they represent the main polyphenol fraction in grape pomace and in several other foods [18]. As the content and profile of phenolic compounds in grape pomace vary according to different factors, such as grape variety, growing conditions, and the specific winemaking process [19], it is important to properly characterize their EPP and NEPP content, and to evaluate how the EPP:NEPP ratio affects the potential health benefits. In the research we report here, we evaluated the effects of three grape pomaces with contrasting EPP and NEPP concentrations and profiles on obesity-related complications. Such a comparison had not previously been performed and it allowed us a better understanding of the benefits associated with EPPs and NEPPs.

## 2. Materials and Methods

### 2.1. Raw Material

Grape pomace from three varieties of grape (Malbec, Garnacha, and Syrah) was collected from different winemakers (La Redonda, Marqués, and Azteca, respectively) located in Querétaro, México, after the wine pressing process. Samples were dried on a tray dryer at 45 °C for 2 days, ground, sieved through a 40-mesh sieve (<425 µm), and stored in hermetic containers at −20 °C protected from light and oxygen until analysis.

### 2.2. Polyphenol Analysis

#### 2.2.1. Total Extractable Polyphenols

The extractable polyphenols were obtained according to the method reported in Pérez-Jiménez et al. [20]. Briefly, each grape pomace sample (100 g) was subjected to two sequential extractions: the first, with methanol:water (50:50 *v*/*v*, pH 2); the second, with acetone:water (70:30 *v*/*v*). The two supernatants were mixed for determination of EPPs (centrifugations were performed at 4000× *g*): extractable phenolic compounds were quantified using the Folin–Ciocalteu method [21], extractable flavonoids according to the procedure reported in Heimler et al. [22], and extractable monomeric anthocyanins by the pH differential method [23]. An additional extraction was performed just with the second solvent combination (acetone:water, 70:30 *v*/*v*) in order to determine the content of extractable proanthocyanidins following the methodology described by Zurita, Díaz-Rubio, and Saura-Calixto [24]. The extraction residue was then used to analyze NEPPs (non-extractable proanthocyanidins (NEPAs) and hydrolyzable polyphenols (HPPs)).

#### 2.2.2. Total Non-Extractable Polyphenols

The resultant dry residue was used to evaluate NEPA content according to the method reported by Zurita, Díaz-Rubio, and Saura-Calixto [24]. HPPs were determined as described by Hartzfeld, Forkner, Hunter, and Hagerman [25], using acid hydrolysis.

#### 2.2.3. Profile of Extractable Polyphenols

The extractable polyphenol profile was assessed in an ultra-performance liquid chromatograph (UPLC) coupled to a quadrupole/time-of-flight mass spectrometer (MS) with an ESI interphase (ESI-QToF MS) (Vion, Waters Co., Milford, MA, USA), as previously reported [26]. The obtained EPP extract (1 mL) was vacuum dried (Speedvac, Savant, Thermo Fisher Scientific, Waltham, MA, USA) and resuspended in 200 µL of methanol and filtered (0.45 µm). Samples were filtered and injected into a BEH Acquity C18 column (2.1 × 100 mm, 1.7 m, Waters Co., Milford, MA, USA) at 35 °C.

Gradient elution was performed with a binary system consisting of (A) water with 0.1% formic acid and (B) acetonitrile with 0.1% formic acid at a flow rate of 0.5 mL/min under gradient conditions of 0 min at 0% B, 2.5 min at 15% B, 10 min at 21% B, 12 min at 90% B, 13 min at 95% B, and 17 min at 0% B. The MS conditions were as follows: capillary voltage, 2.0 kV (ESI−) and 3.5 kV (ESI+); 40 eV of cone voltage; 6 V for low collision energy; 15–45 V for high collision energy; a 120 °C source temperature; 800 L/h of N_2_ at 450 °C as the desolvation gas; 50 L/h cone gas flow. Data were acquired in negative and positive ionization modes (ESI− and ESI+, respectively) within a 100–1800 Da mass range. Polyphenols were identified by analyzing their exact mass (mass error < 5 ppm), isotope distribution, and fragmentation pattern (Appendix A). Calibration curves were constructed with gallic acid (hydroxybenzoic acids), (−)-epicatechin (flavanols), quercetin (flavonols), and cyanidin chloride (anthocyanins).

### 2.3. In Vivo Effects of Grape Pomaces

Forty male Wistar rats (190 ± 5 g) were obtained from the Institute of Neurobiology, UNAM (Querétaro, México). The ethics committee of the Universidad Autónoma de Queretaro (Querétaro, Mexico) approved the experimental protocol (CBQ 18/010). The animals were housed at 25 °C in a 12 h dark/light cycle. After one week of acclimatization, the animals were randomized into five experimental groups of eight rats each: (1) healthy control group fed with a standard diet (STD, Rodent Lab Chow 5001, Purina); (2) obese control group fed a high-fat high-fructose diet (HFFD) consisting of 60% (*w*:*w*) STD diet, 20% (*w*:*w*) pork lard, and 20% (*w*:*w*) fructose, complemented with vitamins and minerals (0.03% *w*:*w*); (3) HFFD + Malbec grape pomace (HFFD + MAL); (4) HFFD + Garnacha grape pomace (HFFD + GAR); and (5) HFFD + Syrah grape pomace (HFFD + SYR). This supplementation consisted of 1 g pomace/kg of body weight/day. Food and water were administered ad libitum for 16 weeks and the animals were weighed weekly.

#### 2.3.1. Food and Energy Intake

Food intake was measured three times per week and is reported as the average value expressed as g/rat/day. Energy intake was calculated as mean food consumption multiplied by dietary metabolizable energy and is reported as kcal/rat/d. The metabolizable energy was calculated by multiplying the intake weight of each nutrient by its metabolizable energy: 4 kcal/g for carbohydrates, 9 kcal/g for fat, and 4 kcal/g for protein.

#### 2.3.2. OGTT Assay

At the end of week 15, fasted animals were subjected to an oral glucose tolerance test (OGTT). Glucose (2 g/kg of body weight) was administered via intragastric gavage, and we obtained blood samples from the tail vein at 0 min (prior to glucose administration) and at 15, 30, 60, and 120 min (after glucose administration) to determine the blood concentration of glucose using a handheld glucose meter (Accu-Chek Performa, Roche, Basel, Switzerland). Results are expressed as mg/dL and the trapezoidal rule was used to calculate the area under the curve (AUC).

#### 2.3.3. Animal Slaughter

After 16 weeks, fasted animals were decapitated by guillotine. Blood samples were collected and centrifuged at 4000× *g* for 15 min to obtain serum samples, which were stored at −80 °C until analysis. Adipose (epididymal, retroperitoneal, and mesenteric) tissue was excised and weighed. The relative weight of each adipose tissue was calculated by dividing the weight of the adipose tissue by body weight. The adiposity index was calculated as the total adipose tissue weight divided by body weight.

#### 2.3.4. Histology Analysis of Mesenteric Adipose Tissue

Mesenteric adipose tissue samples were placed in 10% (*v*:*v*) buffered formalin for histological analysis. Afterwards, the tissues were fixed in paraffin and sectioned into 5 μm sections to be stained in hematoxylin and eosin solution. Finally, photographs were obtained at 40× magnification with an optical microscope (Leica DM500, Heerbrugg, Switzerland). The adipocyte area was determined using ImageJ microscope software to estimate adipocyte hypertrophy. Results are expressed as μm^2^.

#### 2.3.5. Quantification of Serum Lipids

Total cholesterol and triglycerides were quantified in serum samples using enzymatic colorimetric kits (SpinReact, St. Esteve de Bas, Spain).

### 2.4. Statistical Analysis

Results are shown as mean values ± standard deviation. The Shapiro–Wilk test determined data distribution, whereas homoscedasticity was assessed using the Levene test. Means were compared using one-way analysis of variance (ANOVA). The parametric data were analyzed using Tukey’s multiple comparison tests (*p* < 0.05). All statistical analyses were carried out using the JMP software (SAS Institute, Cary, NC, USA). Principal component analysis (PCA) was performed to identify the beneficial health effects associated with each grape pomace. Multivariate analyses were carried out using the statistical software R 3.4.3. Partial least squares discriminant analysis (PLS-DA) was performed to analyze polyphenol compounds associated with each grape pomace using the JMP software.

## 3. Results

### 3.1. Grape Pomace Characterization

Polyphenol characterization was performed in order to ensure the contrasting compositional characteristics of the three grape varieties selected, focusing on their EPP and NEPP content. The main results of this analysis are shown in Table 1. Malbec and Garnacha were the varieties of grape pomace with the highest amount of extractable phenolic compounds and extractable flavonoids, while Malbec pomace had the greatest concentration of extractable monomeric anthocyanins and extractable proanthocyanidins. In contrast, Syrah grape pomace has small amounts of these compounds but the highest content of NEPAs and HPPs, which makes it the richest material in NEPPs. The NEPP:EPP ratio was of 1.48 for Malbec, 1.10 for Garnacha, and 5.76 for Syrah (Table 1).

The hierarchical analysis of individual extractable phenolic compounds in the grape pomaces by UPLC-ESI-QToF-MS, as represented in Figure 1, showed contrasting profiles for each grape pomace variety. The compounds (iso)-rhamnetin, peonidin dihexoside, syringetin hexoside, and malvidin dihexoside differentiate Syrah from Malbec and Garnacha, since they are found in higher concentrations in the former. Meanwhile, Malbec and Garnacha differ from each other since Malbec has a high content of compounds ranging from delphinidin rutinoside to petunidin hexoside, while Garnacha is rich in the rest of the listed compounds.

### 3.2. In Vivo Effects of Grape Pomace Supplementation

The three grape pomace varieties selected for this study were evaluated in a prevention study in rats fed an HFFD; Table 2 shows the results for body weight, food intake, and adipose tissue distribution. The animals fed the HFFD showed increased body weight compared to the STD group (32%, *p* < 0.05). In the supplemented groups (HFFD + MAL, HFFD + GAR, HFFD + SYR), body weight gain was reduced by 15%, 10%, and 11%, respectively, compared to the HFFD group, with this being a significant decrease in all cases (*p* < 0.05). No statistical differences were observed in the food and energy intake between any of the experimental groups.

In agreement with the increased body weight, the HFFD control group showed increased adipose tissue compared to the STD group. Specifically, the adiposity index increased by 168% in the HFFD group compared to the STD group (*p* < 0.05). Supplementation of the HFFD with Malbec, Garnacha, and Syrah grape pomaces decreased the adiposity index by 20%, 14%, and 20%, respectively, compared with the obese (HFFD) group (*p* < 0.05). According to the results shown in Table 2, the reduced accumulation of adipose tissue was most apparent in the mesenteric tissue, with 26% for both the HFFD + MAL and HFFD + GAR groups, and 17% in the HFFD + SYR group, compared to the obese (HFFD) group (*p* < 0.05). This was followed by the retroperitoneal adipose tissue, which was decreased by 26% in the HFFD + GAR group, 15% in the HFFD + SYR, and 14% in the HFFD + MAL group; whereas the epididymal tissue was decreased by 19% in the HFFD + SYR (*p* < 0.05), 16% in the HFFD + MAL (*p* < 0.05), and 9% in the HFFD + GAR (*p* < 0.05) group.

As expected, the obese (HFFD) group showed an increased adipocyte size compared to the healthy group, as observed in Figure 2A (38%, *p* < 0.05), demonstrating the development of adipocyte hypertrophy. No significant differences were observed in adipocyte size compared to the obese (HFFD) group (Figure 2B), despite the previously reported effect of all grape pomace varieties in preventing body weight gain and on the adipose tissue index.

Regarding serum triglycerides (Table 3), the animals fed the HFFD showed increased levels compared to the STD group (+187%, *p* < 0.05). Supplementation of the HFFD with Malbec grape pomace (HFFD + MAL) significantly decreased this parameter by 46% compared with the obese group, whereas a 31% (*p* < 0.05) reduction was found with Garnacha grape pomace supplementation. Supplementation with Syrah grape pomace tended to decrease serum triglycerides by up to 16% compared to the obese control group, but no significant differences were observed. In contrast, administration of the HFFD for 16 weeks did not alter serum cholesterol levels as compared to the STD group (Table 3).

Figure 3 shows the results of the oral glucose tolerance test (OGTT) for the animals fed the HFFD and supplemented with the three varieties of grape pomace. The HFFD group showed increased blood glucose levels after 30 min (*p* < 0.05). Interestingly, the animals fed HFFD supplemented with Malbec, Garnacha, or Syrah grape pomaces showed regulated glucose levels, with values similar to those for the STD group (Figure 3A). This trend was confirmed with the AUC values (Figure 3B).

### 3.3. Integrative Approach to the In Vivo Effects of Grape Pomace Supplementation

PCA multivariate analyses were performed to provide an overview of the physiological modifications caused by grape pomace supplementation. In an initial approach, the main parameters affected by grape pomace supplementation (OGTT results, relative adipose tissue weight, body weight gain, and serum triglycerides) were integrated into a PCA model for the five experimental groups (Figure 4A). The first component (PC1) explained 61% of the variation, where clear discrimination is observed between the STD and HFFD control groups, whereas all the groups supplemented with grape pomace were clustered between these two control groups. The biplot showed that all the physiological parameters were responsible for discrimination of the HFFD group (Figure 4B).

A second PCA model was constructed excluding the control groups, in order to further discriminate between the grape pomace varieties (Figure 4C,D). The PCA model explained an overall variance of 62%, revealing poor discrimination between the three experimental groups for both components (Figure 4C).

Therefore, a third PCA model was constructed with the total EPP and NEPP contents of the three varieties of grape pomace (Figure 4E and Figure 4F). The resultant PCA plot explained 99% of the overall variance. The compounds that most clearly characterized the Malbec and Garnacha varieties were extractable monomeric anthocyanins and extractable proanthocyanidins, followed by extractable phenolic compounds and extractable flavonoids. In contrast, the most characteristic groups of the Syrah grape pomace were NEPAs and HPPs.

Finally, a partial least squares discriminant analysis was constructed to identify the extractable polyphenol discriminants of each grape pomace variety and their possible relation with the health effect on obesity. Figure 5 shows that most of the compounds of the phytochemical profile found on the left side, that is, those that had a higher correlation with lesser metabolic alterations associated with obesity such as serum triglycerides (Figure 5A), body weight gain (Figure 5B), and AUC values from the OGTT (Figure 5C) were the anthocyanins such as malvidin hexoside, malvidin coumaryl hexoside, and malvidin hexoside pyruvic acid, while the compounds least associated with an improvement in obesity-related disorders were flavonoids such as quercetin and kaempferol. In contrast, regarding the values of mesenteric adipose tissue (Figure 5D), the compound that was lesser related to this parameter was Kaempferol. The Malbec variety was characterized by having a higher concentration of anthocyanins such as malvidin acetyl hexoside, and malvidin hexoside. Therefore, these compounds could be partially related to the improvement in the parameters related to the animals that received the treatment of the Malbec variety.

## 4. Discussion

Grape pomace contains EPPs and NEPPs, classes of phytochemicals that have been related to several health benefits. However, the effect of the proportion of each polyphenol fraction in grape pomace on those health benefits has not previously been determined. Therefore, in this study, we aimed to provide insight into this area by performing an in vivo study with three grape pomaces with contrasting compositions.

The NEPP:EPP ratio decreased from Syrah (5.76) to Malbec (1.48) to Garnacha (1.10) pomaces. This variability agrees with reports that the composition of the grape pomace depends on multiple factors such as the geographical origin, ripening time, crop, climate, or the winemaking process [27]. These kinds of factors are also involved in the localization of phenolic compounds in the cell structure, and thus affect the proportion of NEPPs as cell wall-linked phenolic compounds [28]. The specific results obtained here for the main phenolic compounds in grape pomace agree with previous results, with flavan-3-ols ((+)-catechin, (−)-epicatechin, and proanthocyanidins), anthocyanins (malvidin-hexoside, delphinidin-rutinoside, peonidin-rutinoside, and petunidin-rutinoside), and flavonols (quercetin, laricitrin, and syringetin derivatives) being the main phenolic compounds identified in methanol/water extractions from skins and seeds of grape pomace [1]. Similarly, other studies reported malvidin 3-*O*-(6″-*O*-*p*-coumaroyl)-glucoside as one of the major anthocyanins [29], quercetin as the main flavonol [29], and (+)-catechin as the main flavanol [30]. Finally, regarding NEPPs, NEPAs are commonly their major fraction [31], and our results (38–86 mg/g of NEPAs) are in the same range as that reported for grape pomace from the Cencibel variety: 67 mg/g [32].

When the three grape pomaces were tested in an animal model of obesity generated by a high-fat high-fructose diet (HFFD), most of the beneficial effects were exerted by all three varieties. The supplementation with the grape pomaces evaluated in this study reduced the body weight gain induced by the HFFD. Such an effect has been evaluated and associated with extractable proanthocyanidins [33], and the beneficial effect has been associated with the augmented expression of the thermogenic markers UCP1 (uncoupling protein 1) in brown adipose tissue and PGC-1-alpha (peroxisome proliferator-activated receptor 𝛾 co-activator 1 alpha) in inguinal white adipose tissue. Moreover, reduced body weight gain has been associated with decreased expression of lipogenic genes such as FAS (fatty acid synthase) and ME (malic enzyme), as well as LPL (lipoprotein lipase) [34]. Nevertheless, although this effect has commonly been associated with EPP supplementation, we observed an anti-obesogenic effect exerted by Syrah grape pomace, the variety with the lowest EPP proportion, so this beneficial effect may also be exerted by NEPPs.

A similar trend was observed for the effect of grape pomace on relative adipose tissue weight, adiposity index, and glucose tolerance, where both EPPs and NEPPs seemed to contribute to the observed beneficial effects, since the three varieties exerted similar effects. In agreement with this, preclinical and clinical studies with whole grape pomace containing both EPPs and NEPPs had previously found beneficial effects on insulin sensitivity [11,35]. In contrast to these parameters, no modification was observed in the elongation of hypertrophic adipocytes. Again, this agrees with the results of a previous study with grape pomace [13] where the hypertrophic adipocytes did not decrease; nevertheless, the supplementation decreased the development of inflammation of adipose tissue, a parameter not assessed here. Finally, a differential varietal effect was observed for serum triglycerides, which were significantly decreased by Malbec grape pomace. In the case of total cholesterol, the supplementation with the HFFD did not cause a modification in serum levels; this agrees with the results we previously obtained in another study with this kind of diet [36], so it seems hypercholesterolemia is not a regular consequence of this model.

A novelty of this study, besides the simultaneous comparison of the three pomaces with contrasting composition, was our integrative analysis of the results obtained, linking physiological modifications with the profile of phenolic compounds. This showed that the main EPPs associated with an improvement in obesity and its complications were anthocyanins, such as malvidin coumaroyl-hexoside, and malvidin acetyl-hexoside. According to Ohyama et al. [12], a grape seed extract rich in catechins can reduce hyperlipidemia and obesity-related disturbances in mice. In addition, da Costa et al. [37] evaluated an extract obtained from grape skin that was rich in anthocyanins such as peonidin 3-*O*-glucoside, petunidin 3-*O*-glucoside, malvidin 3-*O*-glucoside, and malvidin 3-(6-*O*-*trans*-*p*-coumaroyl)-5-*O*-glucoside; the extract decreased weight gain, dyslipidemia, and insulin resistance in mice fed a high-fat diet. These effects were attributed to the regulation of insulin signaling cascade proteins and increased expression of GLUT4 in adipose tissue and muscle.

## 5. Conclusions

After comparing the effects of three grape pomaces with different NEPP and EPP profiles in an obesity animal model induced by HFFD, the Malbec grape pomace with an NEPP:EPP ratio of 1.48 produced the greatest effect on the control of weight gain, mesenteric adipose tissue weight, adiposity index, glucose homeostasis, and triglycerides in serum. These effects were associated with anthocyanins and flavanols, specifically malvidin acetyl-hexoside, and malvidin hexoside. In contrast, supplementation with Syrah pomace, with an NEPP:EPP ratio of 5.76, produced the same physiological effects except for triglyceride regulation. Therefore, these results suggest that both EPPs and NEPPs improve metabolic alterations related to obesity, but EPPs are more relevant for the prevention of hypertriglyceridemia.

## Figures and Tables

**Figure 1 foods-12-01370-f001:**
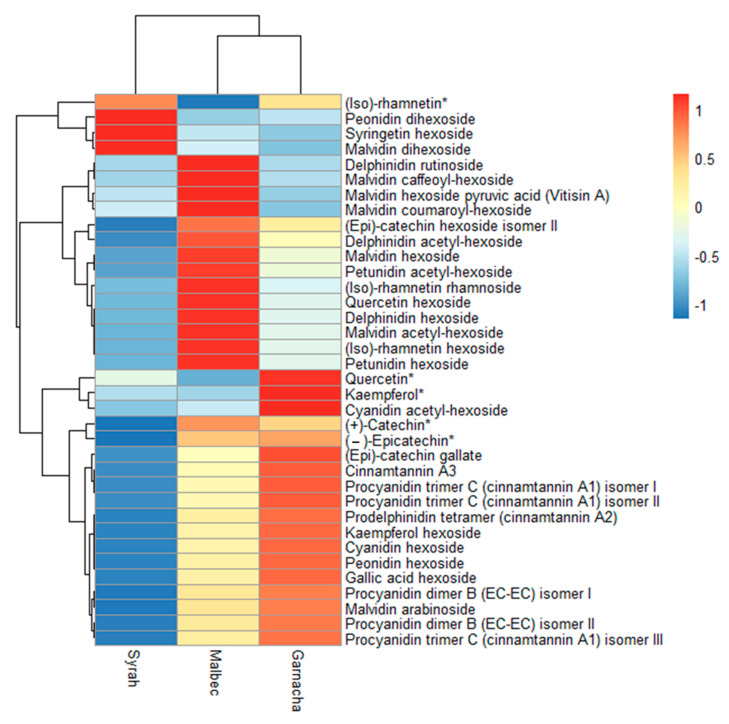
Hierarchical analysis of extractable polyphenol profiles of different varieties of grape pomace. Each colored cell on the map corresponds to a concentration value. * Identification confirmed with commercial standards.

**Figure 2 foods-12-01370-f002:**
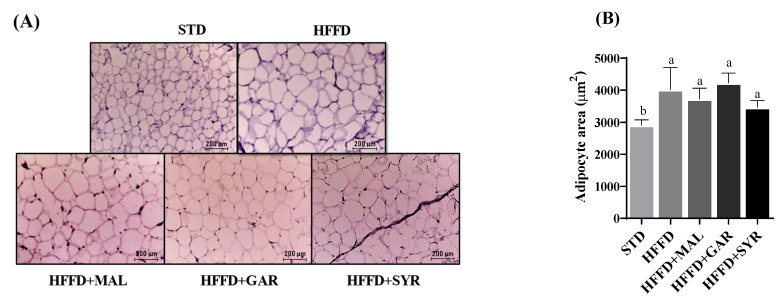
Mesenteric adipose tissue histology analysis and adipocyte area of rats fed a high-fat high-fructose diet supplemented with different varieties of grape pomace. (**A**) Representative histological images of adipose tissue. (**B**) Adipocyte area (µm^2^). Values are reported as mean ± SD (n = 8). Different letters indicate significant (*p* < 0.05) differences according to Tukey’s test. STD; standard diet, HFFD; high-fat high-fructose diet, MAL; Malbec, Gar; Garnacha, SYR; Syrah.

**Figure 3 foods-12-01370-f003:**
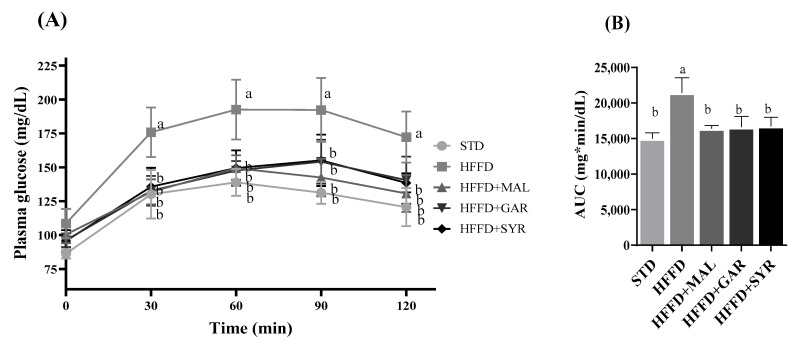
Effect of different varieties of grape pomace on glucose tolerance in rats fed a high-fat high-fructose diet. (**A**) Postprandial blood glucose levels and (**B**) area under the curve (AUC) measured between 0 and 120 min after glucose load. Data are reported as means ± SE for eight animals per group. Statistically significant differences were determined by ANOVA, followed by Tukey’s test (*p* < 0.05). STD; standard diet, HFFD; high-fat high-fructose diet, MAL; Malbec, GAR; Garnacha, SYR; Syrah.

**Figure 4 foods-12-01370-f004:**
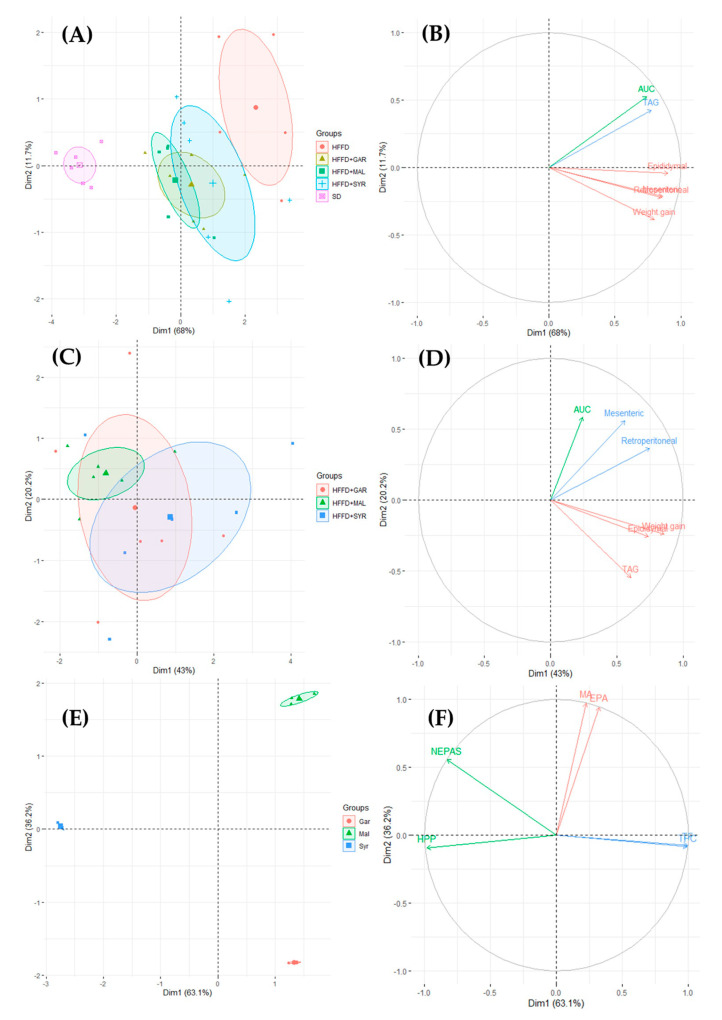
Principal component analysis (PCA) of physiological effects on rats fed a high-fat high-fructose diet supplemented with different varieties of grape pomace (**A**,**B**). Principal component analysis (**C**) and biplot (**D**) of the physiological effects on rats of supplementation of the high-fat high-fructose diet with different varieties of grape pomace. Principal component analysis (**E**) and biplot (**F**) of extractable and non-extractable polyphenol contents in different grape pomace varieties. AUC, area under the curve; EPA, extractable proanthocyanidins; GAR, Garnacha; HFFD, high-fat high-fructose diet; HPP, hydrolyzable polyphenols; MA, extractable monomeric anthocyanins; MAL, Malbec; NEPAs, non-extractable proanthocyanidins; STD, standard diet; SYR, Syrah; TAG, triacylglycerols; TF, total flavonoids; TPC, total phenolics content.

**Figure 5 foods-12-01370-f005:**
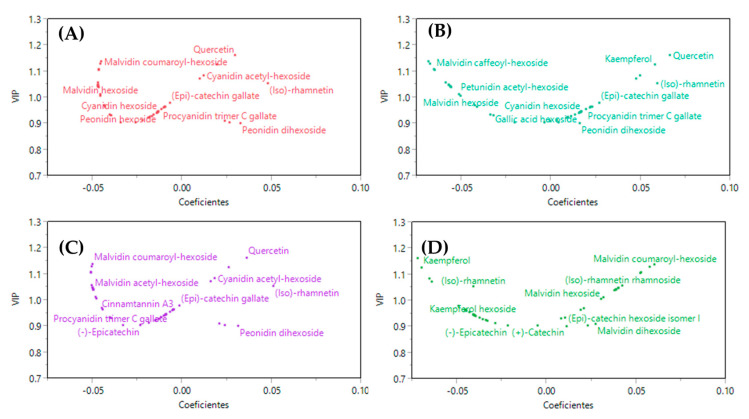
Partial least squares discriminant analysis PLS-DA of the phenolic profile of grape pomace varieties and (**A**) triglycerides in serum, (**B**) weight increase, (**C**) AUC from OGTT, and (**D**) mesenteric adipose tissue.

**Table 1 foods-12-01370-t001:** Extractable and non-extractable polyphenol content of different varieties of grape pomace.

	MAL	GAR	SYR
*Polyphenols in extractable fractions*			
Extractable phenolic compounds	49.4 ± 1.6 ^a^	52.1 ± 1.4 ^a^	16.2 ± 0.4 ^b^
Extractable flavonoids	48.8 ± 1.9 ^a^	51.3 ± 2.2 ^a^	11.0 ± 0.5 ^c^
Extractable monomeric anthocyanins	2.94 ± 0.07 ^a^	0.79 ± 0.03 ^c^	1.47 ± 0.06 ^b^
Extractable proanthocyanidins	7.33 ± 1.30 ^a^	1.88 ± 0.21 ^c^	3.07 ± 0.12 ^b^
Total extractable polyphenols	56.7	54.0	19.3
*Polyphenols in non-extractable fractions*			
Non-extractable proanthocyanidins	64.4 ± 3.1 ^b^	38.8 ± 2.4 ^c^	86.1 ± 4.0 ^a^
Hydrolyzable polyphenols	20.0 ± 0.8 ^b^	20.4 ± 0.7 ^b^	24.9 ± 1.0 ^a^
Total non-extractable polyphenols	84.2	59.2	111.0

Results are expressed in mg/g of grape pomace and given as the average ± SD, n = 3. Means in the same row with different superscript letters differ significantly according to Tukey’s test (*p* < 0.05). MAL, Malbec; GAR, Garnacha; SYR, Syrah.

**Table 2 foods-12-01370-t002:** Effects of a high-fat high-fructose diet supplemented with different varieties of grape pomace on rat growth, food intake, and adipose tissue weight.

	STD Group	HFFD Group	HFFD + MAL	HFFD + GAR	HFFD + SYR
*Growth parameters*					
Initial body weight (g)	192 ± 7 ^a^	194 ± 7 ^a^	196 ± 7 ^a^	190 ± 11 ^a^	197 ± 11 ^a^
Final body weight (g)	540 ± 23 ^c^	716 ± 20 ^a^	612 ± 24 ^b^	645 ± 67 ^ab^	639 ± 42 ^b^
Body weight gain (g)	347 ± 22 ^c^	519 ± 19 ^a^	416 ± 14 ^bc^	455 ± 62 ^ab^	448 ± 42 ^ab^
Food intake (g/rat/day)	28 ± 4 ^a^	24 ± 2 ^a^	26 ± 4 ^a^	26 ± 2 ^a^	25 ± 3 ^a^
Energy intake (kcal/rat/day)	562 ± 32 ^a^	546 ± 48 ^a^	535 ± 80 ^a^	524 ± 67 ^a^	542 ± 86 ^a^
*Adipose tissue, relative weight (g*/*100 g)*					
Epididymal	1.51 ± 0.07 ^d^	3.70 ± 0.28 ^a^	3.11 ± 0.37 ^bc^	3.37 ± 0.25 ^ab^	2.98 ± 0.23 ^bc^
Mesenteric	1.21 ± 0.03 ^d^	4.41 ± 0.16 ^a^	3.27 ± 0.48 ^bc^	3.25 ± 0.35 ^bc^	3.64 ± 0.24 ^b^
Retroperitoneal	0.75 ± 0.10 ^d^	1.84 ± 0.40 ^a^	1.59 ± 0.29 ^ab^	1.36 ± 0.08 ^bc^	1.56 ± 0.13 ^ab^
Adiposity index	4.18 ± 0.36 ^c^	11.21 ± 0.77 ^a^	8.90 ± 1.66 ^b^	9.68 ± 1.26 ^ab^	8.93 ± 1.86 ^b^

Results are expressed as average ± SD, n = 8. Means in the same row with different superscript letters differ significantly according to Tukey’s test (*p* < 0.05). STD; standard diet, HFFD; high-fat high-fructose diet, MAL; Malbec, GAR; Garnacha, SYR; Syrah.

**Table 3 foods-12-01370-t003:** Serum triglyceride and cholesterol levels of rats fed a high-fat high-fructose diet supplemented with different varieties of grape pomace.

	STD	HFFD	HFFD + MAL	HFFD + GAR	HFFD + SYR
*Serum*					
Triglycerides (mg/dL)	58.5 ± 6.9 ^c^	168 ± 30 ^a^	90.4 ± 23.6 ^b^	116 ± 21 ^ab^	141 ± 29 ^a^
Cholesterol (mg/dL)	64.6 ± 8.9 ^a^	69.79 ± 3.4 ^a^	69.8 ± 2.4 ^a^	66.4 ± 4.2 ^a^	68.3 ± 5.2 ^a^

Results are expressed as the average ± SD, n = 8. Means within the same line with different superscript letters differ significantly according to Tukey’s test (*p* < 0.05). STD; standard diet, HFFD; high-fat high-fructose diet, MAL; Malbec, GAR; Garnacha, SYR; Syrah.

## Data Availability

Data are contained within the article.

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
