# Peer review of "Three Varieties of Grape Pomace, with Distinctive Extractable:Non-Extractable Polyphenol Ratios, Differentially Reduce Obesity and Its Complications in Rats Fed a High-Fat High-Fructose Diet"

_foods, 2023, doi:10.3390/foods12071370_

Round 1

Reviewer 1 Report

Dear authors,

The experiments presented in manuscript are well conducted and my recommendations are the following:

-          Lines 135, 136: „g/rat/d”, I suggest,g/rat/day” (I marked with yellow)

-          Line 148: “4000 g”, I suggest, “4000x g” (I marked with yellow)

-          Lines 175, 196, 224: “polyphenol”, I suggest, “polyphenols” (I marked with yellow)

-  Line 196: “different varieties of grape pomace varieties”, I suggest, “different varieties of grape pomace” (I marked with yellow)

-    Lines 200, 252, 278, 291, 292, 332: “malbec;….garnacha; ……syrah”, I suggest, “Malbec;….Garnacha; ……Syrah” (I marked with yellow)

-       Line 232: “15%, 10, and….” , I suggest, “15%, 10%, and…..”(I marked with yellow)

-           Line 240: “20, 14 and 20%” , I suggest, “20%, 14% and 20%”(I marked with yellow)

-          Line 264: “malbec,…..”, I suggest, “Malbec,….” (I marked with yellow)

-          Line 294: “PCA”, I suggest, “Principal Component Analysis (PCA)” (I marked with yellow)

-          Line 324, Figure 4: “Principal Component Analysis”, I suggest, “Principal Component Analysis (PCA)” (I marked with yellow)

-          Line 333: “Partial Least Squares Discriminant Analysis”, I suggest, “Partial Least Squares Discriminant Analysis (PLS-DA)” (I marked with yellow)

-  Line 346: “Partial Least Squares Discriminant Analysis PLS-DA”, I suggest, “Partial Least Squares Discriminant Analysis (PLS-DA)” (I marked with yellow)

-      Line 408: “NEPP.EPP”, I suggest, “NEPP : EPP” (I marked with yellow)

-   Line 503: “Vitis vinifera” , I suggest Italic font,Vitis vinifera (I marked with yellow)

-    I suggest to use the same formulation throughout all the text, or “(p < 0.05)”, or (P < 0.05)  (I marked with yellow)

-    At Materials and Methods (criteria 2.2.3 and all 2.3), I reccommend if is possible, to mention the references for each specific used method, according to……

Thank you!

Reviewer 2 Report

Comments to the Author

Minor Revisions

1.     Whole document: Language use should be improved; the text has numerous spelling and grammatical mistakes. English editing is needed, and the authors are advised to have the whole manuscript checked by a native English speaker. Authors should scan the entire manuscript for minor punctuation and grammatical errors.

2.     Abbreviations: should be defined at 1st mention, so please write the full name when first mentioned; then you can use the abbreviation later in the article.

3.     Abstract:

              ·         The author wrote in line 28: The Serum triglycerides significantly decreased with Malbec by 53% and Garnacha by 30% compared to the HFFD group, but not significantly with Syrah.

Again, in line 266, they wrote: Regarding serum triglycerides (Table 3), The supplementation of the HFFD with Malbec grape pomace significantly decreased this parameter by 46%, whereas a 31% reduction was found with Garnacha grape pomace supplementation. Even though the supplementation with Syrah grape pomace decreased serum triglycerides by 16% as compared to the obese control group, no significant differences were observed.

Please revise the percentages…….

4.     Introductions:

              ·         It is very poor, and more in-depth information and detailed background and scientific context focus on some points are needed to be added to the study, such as about:

A.    Triglcride as the main storage form of fats within the body.

B.    Obesity and its complications.

5.     Materials and Methods:

              ·         Why did you not try different doses to demonstrate your hypothesis?

              ·         Authors should add the make and country of all instruments used in the study.

              ·         Method 2.2 - Mention the g value for centrifugation.

              ·         Line 120: In vivo effects of grape pomaces, the authors must mention the concertation of grape pomaces used in the experiments.

              ·         Why did you not try different doses of the three pomaces to demonstrate your hypothesis?

              ·         Why did the authors not study the effect of these three pomaces on the pancreas and insulin secretion to detect the improvement?

              ·         Why did the authors not study the regulation of insulin signaling cascade proteins and expression of GLUT4 in adipose tissue and muscle?

              ·         Why did the authors not consider the duration of diet intake in their study, although this is very important? Do you think the results can change significantly depending on the time of a high-fat diet intake?

              ·         Why did you not perform an Immunohistochemistry study to support your results?

6.     Results:

              ·         Table 1: What are a, b, and c written in tables 1, 2, and 3? Statistical significance should be mentioned below the table.

              ·         Line 223: Remove the 3.2 In v

              ·         Histopathology results were unclear, the mesenteric adipose tissue histology analysis and the adipocyte area of rats were not apparent, and even the improvement was indefinite. Please add clearer ones to support your hypothesis in Figure 2A, or the resolution of this figure needs to be improved.

              ·         The title of figure 2B needs revision.

              ·         Table 3, Line 271: the authors wrote that the administration of the HFFD for sixteen weeks did not alter serum cholesterol levels compared to the STD group (Table 3). Although the author wrote that Means within the same line with different superscript letters differ significantly, how do you explain this? Please explain what a, b, and c are. Statistical significance should be mentioned in the table.

7.     Discussion

           ·         The authors ignore thoroughly discussing that serum cholesterol levels did not alter as compared to the STD.

8.     Conclusion: is not supported by the results of this manuscript. The authors should rephrase their conclusion to align with their findings.

9.     Acknowledgments: We would like to thank Laura González-Dávalos for his technical support; please correct this to her.

Best Regards

Reviewer 3 Report

The paper Titled: “Three varieties of grape pomace, with distinctive relationship between extractable and non-extractable polyphenols, differentially improve the obesity and its complications of rats fed with a high‐fat ‐fructose diet”, could represent a valid work, but I have many doubts about it.

First of all, in my opinion it is a job that evaluates more the medical aspect than to the Novel Foods or Food Additives, naturally it is my idea. Moreover, in the text there are many errors, both related to the formatting of the paper (check it) and to the English form.

-        I advise to the authors to review the title, perhaps synthesizing it

-        Abstract: see the guidelines of the journal, you should use about 200 words maximum; and I advise to authors to organize better this section.

-        2.2.1 Total extractable polyphenols:

lines 87-94: You write that….”the two supernatants were mixed and after analyzed..”, but after you write that… “The supernatant obtained from the second extraction”…. it is not very clear what you have done, I ask you to write this part better.

-        Pag. 3, line 107. enter the specifications of the column used (mod., brand, country).

-        Fig.1. pay attention to the number of pages and other errors

-        Fig, 5 is not very readable.

Round 2

Reviewer 3 Report

The authors have corrected and improved the article, responding to all the doubts I posed. 

I think that the article can now be considered for publication in this Journal. 

Author Response

Thanks for your positive comments about the article. No modification was added to the manusicrpt, since no additional suggestion was performed.